

# The relationship between (sub)tropical climates and the incidence of COVID-19

David Prata[1,*], Waldecy Rodrigues[1,*],
Paulo Henrique De Souza Bermejo[2], Marina Moreira[2],
Wainesten Camargo[1], Marcelo Lisboa[1], Geovane Rossone Reis[1] and
Humberto Xavier de Araujo[1]

[1] Institute of Regional Development, Graduate Program of Computational Modelling,
Federal University of Tocantins, Palmas, Brazil
[2] Research and Development Center for Public Sector Excellence and Transformation (NExT) of
the Department of Administration, Federal University of Brasilia, Brazil
* These authors contributed equally to this work.

Corresponding author
David Prata, ddnprata@uft.edu.br

## ABSTRACT

This work explores (non)linear associations between relative humidity and temperature and the incidence of COVID-19 among 27 Brazilian state capital cities in (sub)tropical climates, measured daily from summer through winter. Previous works analyses have shown that SARS-CoV-2, the virus that causes COVID-19, finds stability by striking a certain balance between relative humidity and temperature, which indicates the possibility of surface contact transmission. The question remains whether seasonal changes associated with climatic fluctuations might actively influence virus survival. Correlations between climatic variables and infectivity rates of SARS-CoV-2 were applied by the use of a Generalized Additive Model (GAM) and the Locally Estimated Scatterplot Smoothing LOESS nonparametric model. Tropical climates allow for more frequent outdoor human interaction, making such areas ideal for studies on the natural transmission of the virus. Outcomes revealed an inverse relationship between subtropical and tropical climates for the spread of the novel coronavirus and temperature, suggesting a sensitivity behavior to climates zones. Each 1 °C rise of the daily temperature mean correlated with a $-11.76\%$ ($t = -5.71$, $p < 0.0001$) decrease and a 5.66% ($t = 5.68$, $p < 0.0001$) increase in the incidence of COVID-19 for subtropical and tropical climates, respectively.

## INTRODUCTION

The novel coronavirus (COVID-19), is caused by the severe acute respiratory syndrome coronavirus 2 (SARS-CoV-2), and due to its high contiguousness and widespread, it has officially been declared a pandemic by the World Health Organization (WHO) on March 11, 2020. As of November 18th, 2020, more than 1.3 mi deaths by COVID-19 were confirmed based on 55 mi confirmed cases in 219 countries (*WHO, 2020*).

In Brazil, the COVID-19 pandemic began on February 26th, 2020, in São Paulo (subtropical city) (*Da Saúde, 2020*). After 43 days, it already killed more people than H1N1, dengue, and measles combined throughout 2019 (*Costa, 2020*). In April, the health

system of the city of Manaus (capital of Amazonas state), collapsed, followed by the funerary system (*Campbell, 2020*).

The spread of the virus from temperate to tropical (*Auler et al., 2020*) regions subverted the expectation (*Bukhari & Jameel, 2020*; *Liu et al., 2020*; *Xie & Zhu, 2020*; *Zhu et al., 2020*; *Prata, Rodrigues & Bermejo, 2020*; *Núñez-Delgado, 2020*; *Yip et al., 2007*; *Thai et al., 2015*; *Ng, Basta & Cowling, 2014*; *Lowen & Steel, 2014*; *Bi, Wang & Hiller, 2007*; *Barreca & Shimshack, 2012*; *Moriyama & Ichinohe, 2019*; *Casanova et al., 2010*; *Wang, Goggins & Chan, 2018*; *Xu et al., 2020*) that the virus would not spread as efficiently in warmer climates. Debunking this hypothesis, studies as of *Kratzel et al. (2020)* have indicated that Sars-CoV-2 incubation has the highest predicted half-life at 30 °C, much higher than originally believed. Other experiments have shown that the survival of Sars-CoV-2 is roughly five times more likely in humid conditions than in dry conditions (*Bhardwaj & Agrawal, 2020*). As of mid-April 2020, the world has seen a surge in the number of cases in regions with mean temperatures above 18 °C, like Brazil.

Evidence suggests that Sars-CoV-2 finds stability in striking a balance between relative humidity and temperature, which indicates the possibility of surface contact transmission. The question remains whether seasonal changes associated with climatic fluctuations might actively influence virus stability and survival. Surface contact transmission evaluations have revealed that this method has a significant impact on virus transmission (*Casanova et al., 2010*; *Van Doremalen, Bushmaker & Munster, 2013*). Temperature and humidity variations have also been proven to influence SARS-CoV-2 stability and survival on surfaces (*Duan et al., 2003*). Among other factors, depending on temperature and humidity, the infectious capacity of the virus can persist on surfaces for anywhere from a few hours to several days (*Van Doremalen, Bushmaker & Munster, 2013*).

Regardless the great number of studies modeling the effects of meteorological conditions on COVID-19 transmission conducted worldwide, findings concerning the relationship of meteorological conditions and COVID-19 transmissibility are still controversial (*Auler et al., 2020*).

This study may help health policymakers apply knowledge about seasonality, infection prevention, and control to prevent local transmission and slow the spread of the novel coronavirus to (sub)tropical climates (*Bedford et al., 2020*). Specially, this research pursue the following question: seasonal changes associated with climatic factors fluctuations as defined by Köppen climate types might actively influence virus survival? Brazil's considerable area has allowed researchers to study different climates within one nation. Warmer climates allow for more frequent outdoor human interaction, making such places ideal for studies on the natural transmission of the virus. The research design is presented in Fig. 1.

# MATERIALS AND METHODS

## Study area

The study included 27 cities, all state capitals of Brazil, covering longitudes from 34° 51′ 40″ W to 67° 48′ 27″ W and latitudes from 8° 45′ 43″ N to 30° 1′ 40″ S. Figure 1 shows the Köppen climate types of Brazil (adapted by the authors). In Brazil, 93% of the

**Fact 1:** A great number of studies modeling the effects of meteorological conditions on COVID-19 transmission were conducted worldwide.

**Fact 2**: Findings regarding the relationship of meteorological conditions and COVID-19 transmissibility are still controversial.

**Fact 3:** Previous laboratory analyses have shown that SARS-CoV-2, the virus that causes COVID-19, finds stability by striking a certain balance between relative humidity and temperature.

**Hypothesis:** Seasonal changes associated with climatic factors fluctuations, as defined by Köppen might actively influence virus survival?

**Climatic Factors:** Three characters, where the first indicates the climate zone, e.g. tropical and subtropical, and is defined by temperature and rainfall, symbolize the Köppen climate types. The second considers the rainfall distribution, and the third is the seasonal temperature variation.

**Statistical Methods:**

GAM and LOESS

**Study Case:** Temperature and relative humidity of 27 state capital cities of Brazil, from summer to winter.

**Findings:** Revealed an inverse relationship between subtropical and tropical climates for the spread of the novel coronavirus and temperature, suggesting a sensitivity behavior to climates zones, and perhaps hemispheres. Indeed, Köppen climate types do not follow latitudinal orders to define climate zones

**Figure 1** **Schematic research design.** Facts, hypothesis, methods, and findings.

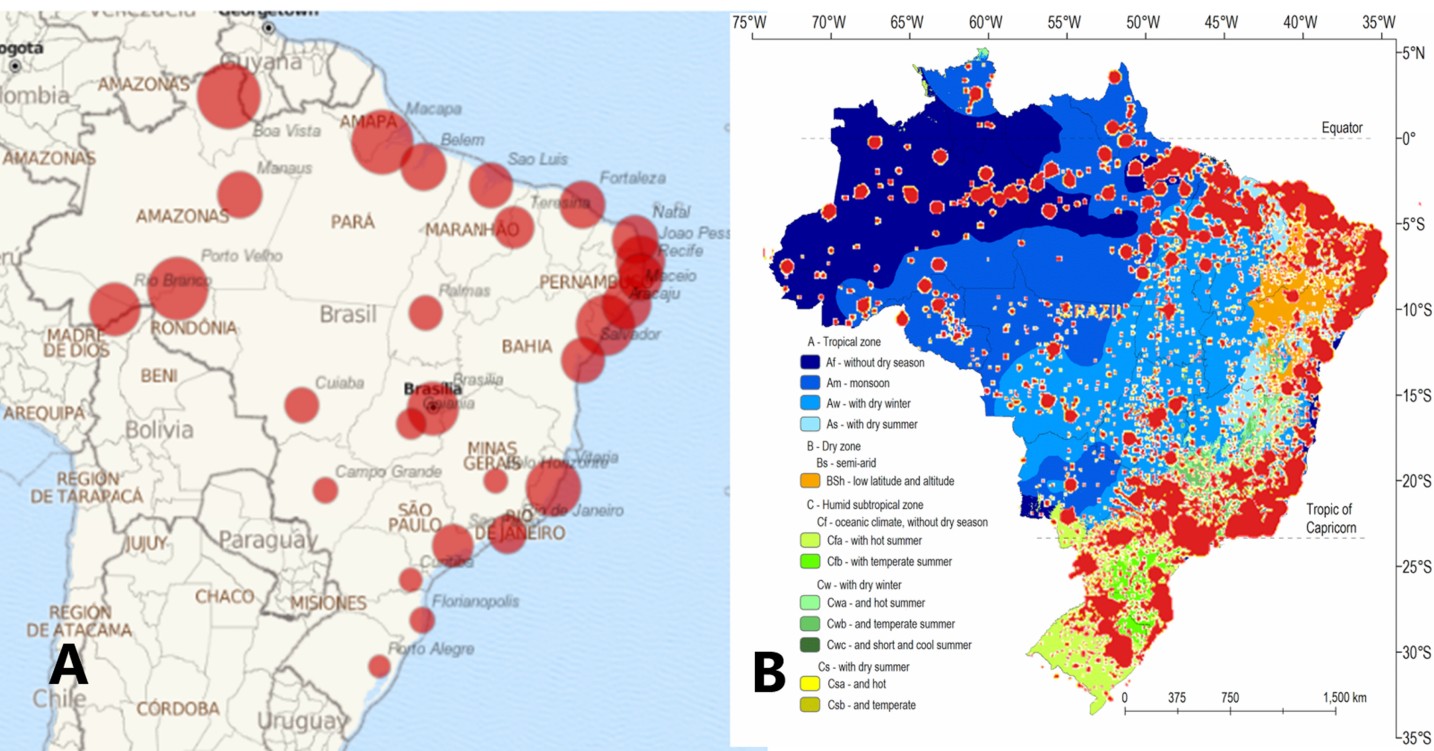

**Figure 2 COVID-19 in state capital cities of Brazil.** On the left, COVID-19 incidence by state capitals. On the right, Köppen Climate Types of Brazil (From *Alvares et al., 2013*; adapted by Authors) and confirmed cases of COVID-19.           

landmass is in the Southern Hemisphere (below Equator Line), and the remaining 7% is in the Northern Hemisphere. This puts all but the southernmost part of the country in the planet's tropical zone. The subtropical region, which lies below the Tropic of Capricorn, contains 6.76% of the population. In this work, the term (sub)tropical means both subtropical and tropical Brazilian climates, respectively. In Fig. 2 (*Alvares et al., 2013*), tropical climates like that of Amazonas appear in dark blue, while red dots signify COVID-19 outbreaks, which are especially concentrated in coastal towns.

## Data collection

The study population is the total cumulative daily number of confirmed cases of COVID-19 in the 27 state capitals, as officially reported by the Ministry of Health of Brazil from February 26th to July 2nd, 2020. This work focused on the capital cities because of Brazil's vast territory and the lack of climatic data for the interior cities of the Brazilian states. Daily meteorological means were collected from the National Institute of Meteorology (INMET) for the period of study. The estimated 2019 population was derived from the Brazilian Institute of Geography and Statistics (IBGE), the official provider of geographic and statistical information in Brazil. The classification in subtropical and tropical climates follows Köppen climate types. Three characters, where the first indicates

the climate zone and is defined by temperature and rainfall, symbolize the Köppen climate types. The second considers the rainfall distribution, and the third is the seasonal temperature variation.

## Statistical analysis

A descriptive analysis was performed, with numerical variables described using means, standard deviation, and distribution. A Generalized Additive Model (GAM) and Locally Estimated Scatterplot Smoothing (LOESS) nonparametric models were used to calculate the relationships between environmental factors and the logarithm of the number of daily cumulative confirmed cases (lgN) to fit equations and splines. GAM fits generalized additive models for parametric and nonparametric regression and smoothing to explore linear and nonlinear weather effects and health outcomes (*Liu et al., 2020*; *Wu et al., 2018*). GAM can be useful to explore linear and nonlinear weather effects and health outcomes (*Zhu et al., 2020*). Although GAM and LOESS use the same statistical technique loess by smoothing out the data in local neighborhoods, LOESS is able to capture significant smoothing features because the local regions are treated independently. While GAM imposes additive structure, requiring that cross sections of the fitted surface always have the same shape. The use of both techniques together are more suitable for exploring the data and visualizing the relationship between the dependent variable and the independent variables than traditional linear models. In some cases, LOESS can capture essentially smoothing featureless of GAM. The model equations were defined as follows:

$$(y_{it}) = \beta_0 + \beta_1(T_{it}) + s(T_{it}) + \varepsilon_{it} \tag{1}$$

$$(y_{it}) = \beta_0 + \beta_1(rh_{it}) + s(rh_{it}) + \varepsilon_{it} \tag{2}$$

$$(y_{it}) = \beta_0 + \beta_1(T_{it}) + \beta_2(rh_{it}) + s(T_{it}) + s(rh_{it}) + \varepsilon_{it} \tag{3}$$

These equations from (1) to (3) show the daily cumulative COVID-19 counts per 100 K inhabitants $(y_{it})$ in capital city $i$ on day $t$, considering the corresponding daily mean temperature $(T_{it})$ and daily mean relative humidity $(rh_{it})$. $\beta_0$ is the intercept, $\beta_1$ and $\beta_2$ are the parameters of linear $(T_{it})$ or $(rh_{it})$, and $s(\cdot)$ denotes a spline function with a maximum of two degrees of freedom to avoid overfitting (*Liu et al., 2020*).

In Eq. (1), daily cumulative COVID-19 counts per 100 K inhabitants $(y_{it})$ is calculated considering just the daily mean temperature $(T_{it})$. Whereas, in Eq. (2), daily cumulative COVID-19 counts per 100 K inhabitants $(y_{it})$ is calculated considering just the daily mean relative humidity $(rh_{it})$. Finally, in Eq. (3), daily cumulative COVID-19 counts per 100 K inhabitants $(y_{it})$ is calculated considering daily mean temperature $(T_{it})$ and the daily mean relative humidity $(rh_{it})$ simultaneously.

The GAM and Loess models were built in SAS™ software, with two-sided tests, and $p < 0.05$ was considered statistically significant.

**Table 1 Descriptive statistics for the daily cumulative confirmed cases of COVID-19.** Descriptive statistics for the daily cumulative confirmed cases of COVID-19 since the first outbreak in each city, and meteorological variables.

| Variable | N | Mean | Std Dev | Minimum | Maximum |
|---|---|---|---|---|---|
| Population | 2,990 | 1,953,711 | 2,560,825 | 299,127 | 12,252,023 |
| Cases (cumulative) | 2,990 | 5,738.03 | 13,015.10 | 1.00 | 129,328 |
| Countdays | 2,990 | 56 | 32.27 | 1.00 | 129.00 |
| T (Daily) | 2,578 | 23.85 | 3.95 | 6.57 | 32.53 |
| Rh (Daily) | 2,544 | 74.45 | 11.33 | 39.87 | 97.37 |

Note:
T, daily mean temperature; Rh, daily mean relative humidity; N, Number of observations.

**Table 2 Pearson correlation coefficients for the incidence of COVID-19.** Pearson correlation coefficients between the incidence of COVID-19 and daily temperature and relative humidity, across all cities and days.

| Variable | Casesh | T(D) | Rh(D) |
|---|---|---|---|
| Casesh (N/hab) | 1.000 | L/NP | L/NP |
| T (daily) (°C) | 0.15437* | 1.000 | L/NP |
| Rh (daily) (%) | 0.25090* | 0.24334* | 1.000 |

Notes:
* $p < 0.0001$.
L, Significant linear correlation; NP, Significant Spearman's non-parametric correlation.

## RESULTS

### Descriptive analysis

Between February 26th and July 2nd, 2020 (129 days), 2,990 (N) observations were collected. Table 1 summarizes the daily cumulative confirmed cases and the meteorological variables. The mean estimated population for the 27 capital cities in 2019 was 1,953,711 inhabitants. The respective daily means for relative humidity and temperature were 74.45% and 23.85 °C. The daily mean of cumulative confirmed cases was 5,738.

Table 2 displays the total number of confirmed cases of COVID-19 per 100 K habitants (incidence), which includes the Pearson correlation coefficients in the cases. The incidence of COVID-19 registered significantly positive Pearson correlations with a daily mean temperature of ($r_p = 0.15437$, $p < 0.0001$) and a daily mean relative humidity of ($r_p = 0.25090$, $p < 0.0001$). However, the nonparametric Spearman's correlation was negative to daily mean temperature of ($r_s = -0.08075$, $p = 0.0009$) and a positive daily mean relative humidity of ($r_s = 0.20703$, $p < 0.0001$). The negative nonparametric result of Spearman's correlation has also given the intuition to explore linear and nonlinear weather effects on the analysis.

### Dose–response relationship

The dose–response relationships among temperature and humidity for (sub)tropical climates are shown in Fig. 3, where nonparametric and linear trends were analyzed by

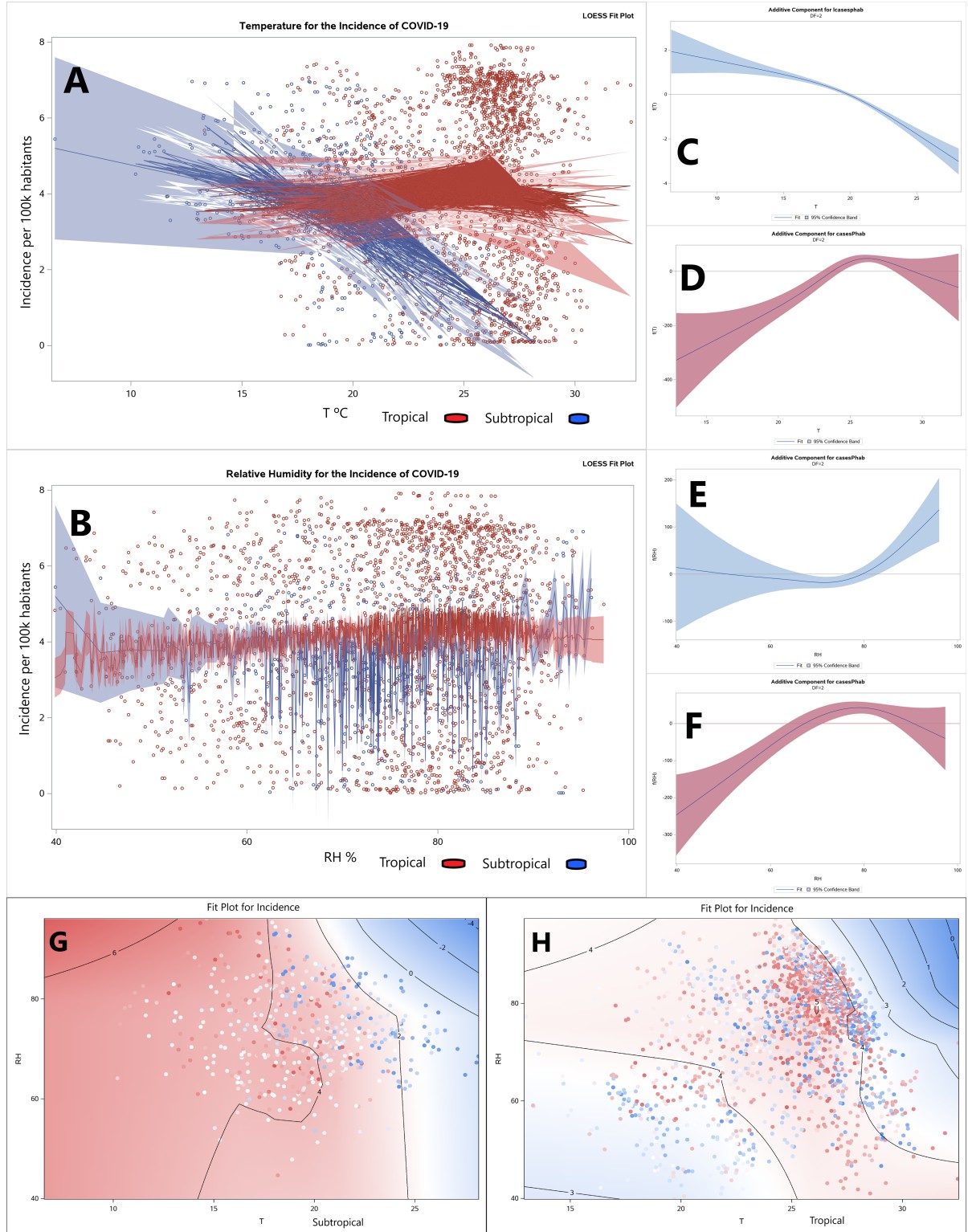

**Figure 3 Dose–response relationships for the incidence of COVID-19.** The dose–response relationships of temperature and relative humidity by the LOESS (A and B) and GAM (C–F) regression models, and the LOESS bivariate smoothing (G and H).

GAM models considering Eqs. (1) and (2) and the LOESS model considering Eq. (3), with all of these equations stated in "Statistical Analysis".

A positive correlation emerged between daily mean temperature and incidence of COVID-19 in tropical regions, while the inverse was true in subtropical regions. The dose–response relationships of temperature (Fig. 3A) and relative humidity (Fig. 3B) by the LOESS regression model. The dose–response relationships of daily means of temperature (Fig. 3C subtropical, Fig. 3D tropical) and relative humidity (Fig. 3E subtropical, Fig. 3F tropical) by the univariate GAM model. The bivariate smoothing of the daily means of temperature (Fig. 3G) and relative humidity (Fig. 3H) by the LOESS model.

The incidence of COVID-19 cases is correlated for each degree Celsius in temperature and each percentage point in relative humidity. Every 1 °C rise of the daily temperature mean correlated with a −11.76% ($t = -5.71$, $p < 0.0001$) decrease and a 5.66% ($t = 5.68$, $p < 0.0001$) increase in the incidence of COVID-19 for (sub)tropical climates, respectively. And, a positive correlation emerged between daily mean humidity and incidence of COVID-19 for both (sub)tropical regions. Each 1% rise of the daily relative humidity mean was associated with a 1.89% ($t = 2.35$, $p = 0.0193$) and a 1.17% ($t = 4.60$, $p < 0.0001$) increase of COVID-19 for (sub)tropical climates, respectively.

## Sensitivity analysis: the lag of days

The splines of the lags of 14 and 21 days for the daily temperature and relative humidity means retain the same linearity in Fig. 3 above. For this reason, the study here plotted a LOESS fit to capture distinct smoothing features of the regression line (Fig. 4).

The concentration of cases for subtropical temperatures varies from approximately 15 °C to 25 °C. In this range, cases decrease as temperature rises (Figs. 4A and 4B), for lags of 14 and 21 days. In contrast, the tropical climate experiences a case increase as temperature rises, with cases peaking between 24 °C and 28 °C (Figs. 4C and 4D), for lags of 14 and 21 days.

For relative humidity, in general, incidence grows from 40% onwards in both subtropical and tropical climates, as shown in Figs. 4E–4H, for lags of 14 and 21 days.

## Sensitivity analysis: seasonal change

During this data analysis, Brazil transitioned from summer to winter. Therefore, it follows that since the first outbreak in the summer month of February, the temperature had decreased day by day as winter arrived, while cases increased (Fig. 5A).

The biased slope of temperature was −0.017620 ($F = 54.97$, $p < 0.0001$), which means that for each day along the $x$-axis, the daily temperature decreased −0.017620 units. To offset bias in this analysis, the temperature data was adjusted as follows:

$$Td_{it} = T_{it} + ((CountDays_i - 1) * 0.017620) \qquad (4)$$

$Td_{it}$ is the adjusted temperature in the capital city $i$ on day $t$. $T_{it}$ is the observed temperature in the capital city $i$ on day $t$. The variable $CountDays_i$ are the days since the first outbreak in city $i$.

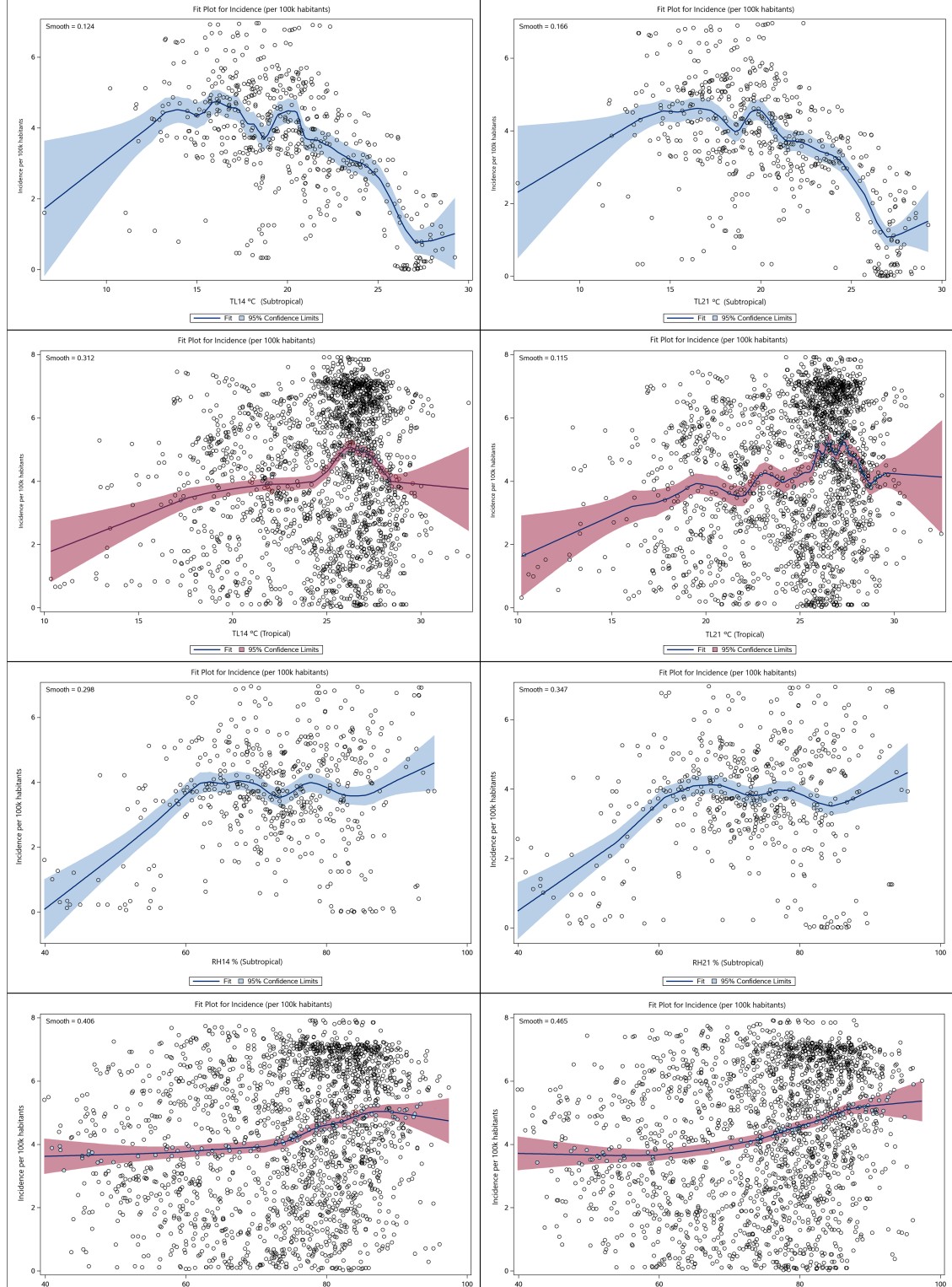

**Figure 4 The lag of days for the dose-response relationship.** The dose–response relationship of temperature and relative humidity for lags of 14 and 21 days by (sub)tropical regions.

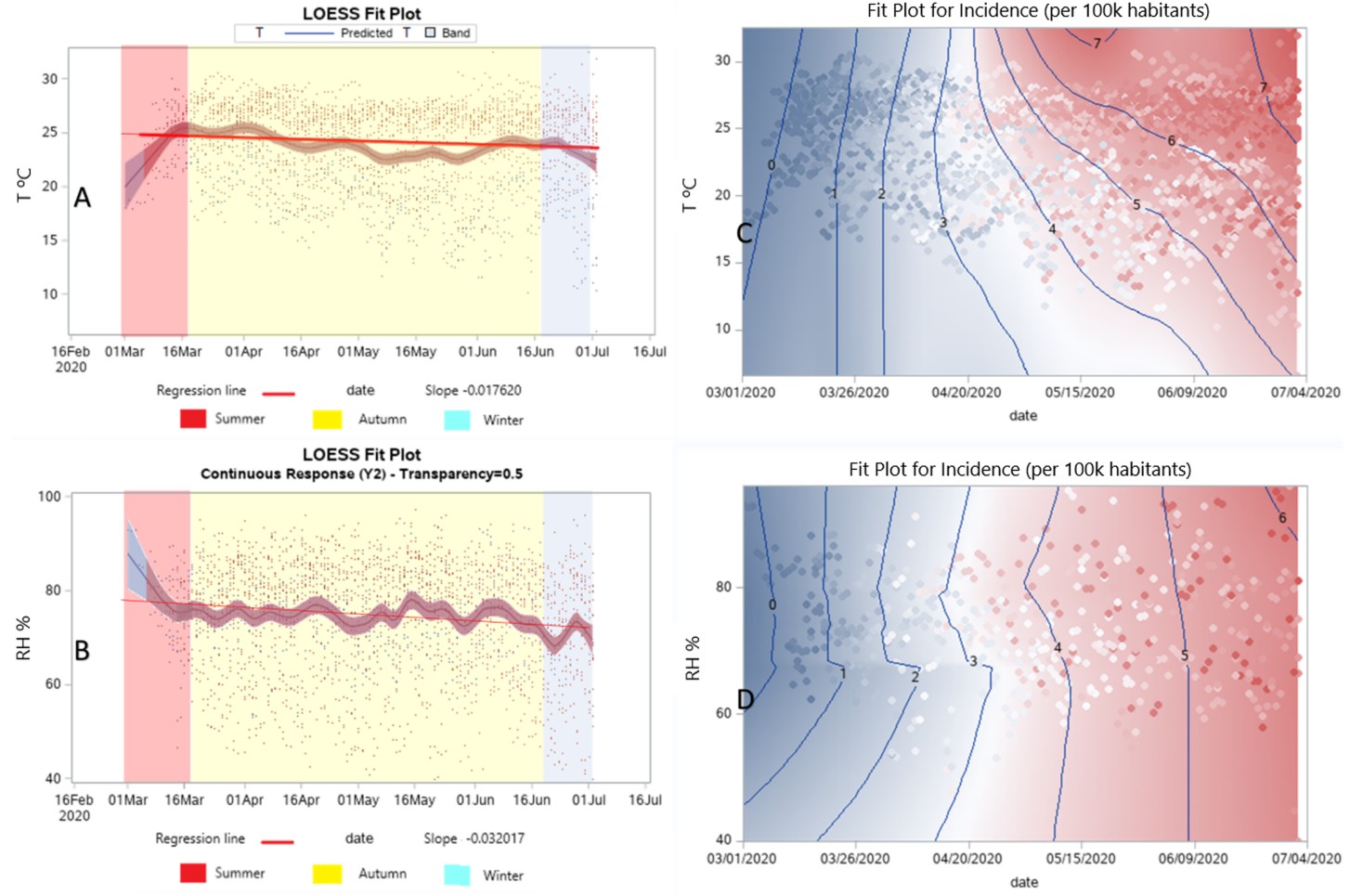

**Figure 5 Daily mean temperature and relative humidity by time across seasons.** The regression line for mean daily temperature (A) and relative humidity (B) along time; and the bivariate smoothing of temperature (C) and relative humidity (D) for the spread of the incidence of COVID-19 over time.

The biased slope of relative humidity (Fig. 5B) was −0.032017 ($F$ = 21.47, $p$ < 0.0001), and the relative humidity data was adjusted as follows:

$$rhd_{it} = rh_{it} + ((CountDays_i − 1) * 0.032017) \tag{5}$$

$rhd$ is the adjusted relative humidity in the capital city $i$ on day $t$. $rh$ is the daily observed relative humidity in the capital city $i$ on day $t$. The variable count days are the days since the first outbreak in city $i$.

In Figs. 5C and 5D, the bivariate smoothing of temperature and relative humidity can evidence the spread of the incidence of COVID-19 over time.

After adjusting the biased slopes of temperature and relative humidity, the slopes changed to a relatively flat regression lines of −0.0002723 ($F$ = 0.01, $p$ = 0.9078), and −0.0005267 ($F$ = 0.01, $p$ = 0.9394), respectively. The inverse relationship of temperature becomes even more clearly on Figs. 6A and 6B by the LOESS regression model, and in Figs. 6C and 6D by the univariate GAM model.

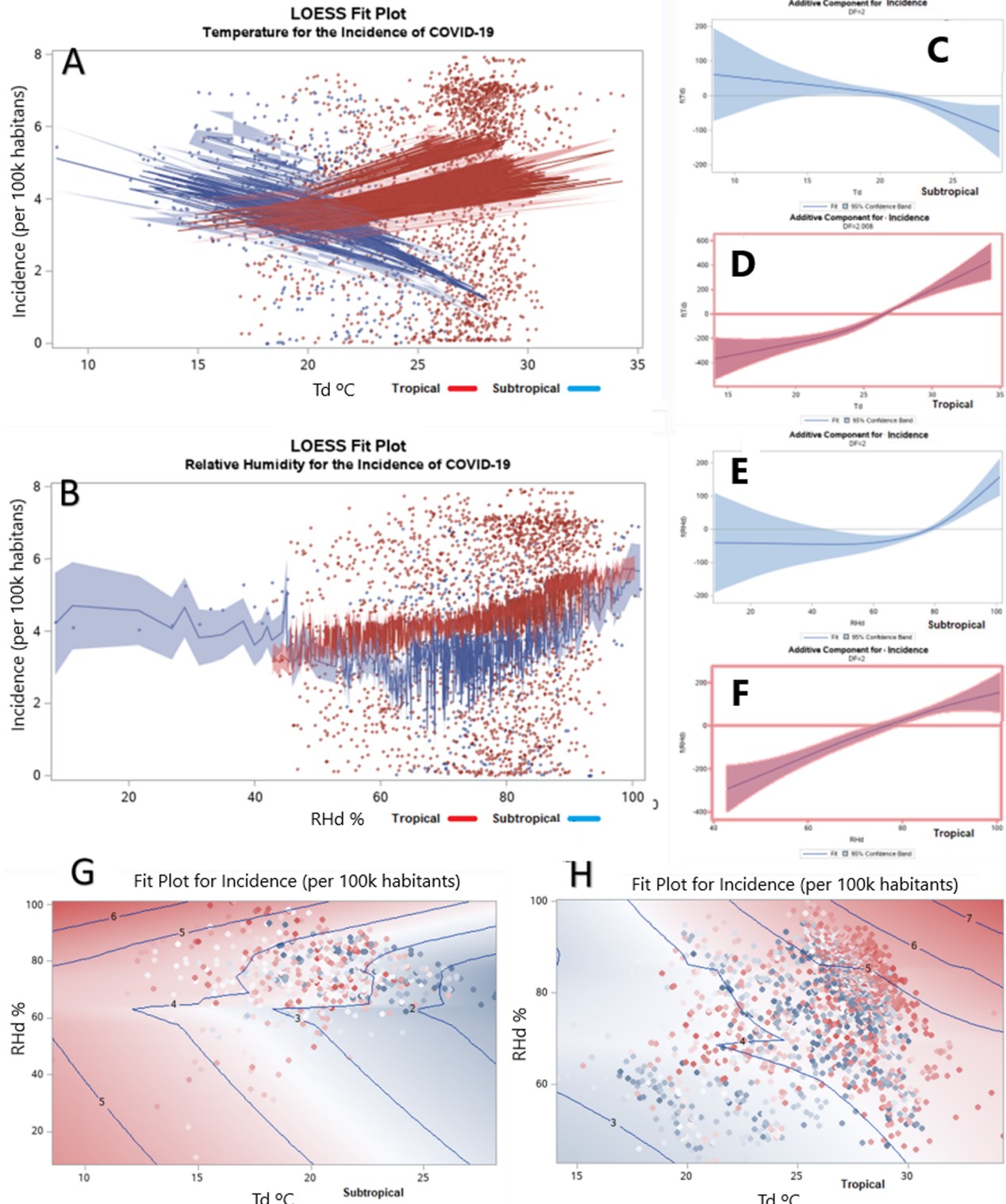

**Figure 6 Dose-response relationships regarding seasonal change.** The dose–response relationships of temperature and relative humidity by the GAM and LOESS (A and B) and GAM (C–F) regression models; the bivariate smoothing (G and H) of the daily means of temperature and relative humidity by the LOESS model.

**Table 3 The effects of a 1 °C T increase in the incidence of COVID-19 confirmed cases.** The effects of a 1 °C increase in daily means of temperature on the incidence of COVID-19 confirmed cases.

**Daily temperature mean from 6.57 °C to 32.53 °C**

|  | Percentage change (%) | | t-value | | p | |
| --- | --- | --- | --- | --- | --- | --- |
|  | SUB | TROP | SUB | TROP | SUB | TROP |
| Incidence | −11.76 | 5.66 | −5.71 | 5.68 | <0.0001 | <0.0001 |
| Incidence L0-07 | −6.18 | 6.26 | −2.94 | 6.29 | 0.0035 | <0.0001 |
| Incidence L0-14 | −8.95 | 5.83 | −4.25 | 5.84 | <0.0001 | <0.0001 |
| Incidence adjusted | −6.89 | 12.14 | −2.99 | 12.35 | 0.0029 | <0.0001 |

**Table 4 The effects of a 1% RH increase in the incidence of COVID-19 confirmed cases.** The effects of a 1% increase in mean relative daily humidity on the incidence of COVID-19 confirmed cases.

**Mean relative daily humidity from 39.87% to 97.37%**

|  | Percentage change (%) | | t-value | | p | |
| --- | --- | --- | --- | --- | --- | --- |
|  | SUB | TROP | SUB | TROP | SUB | TROP |
| Incidence | 1.89 | 1.17 | 2.35 | 4.60 | 0.0193 | <0.0001 |
| Incidence L0-07 | 1.25* | 1.65 | 1.51 | 6.46 | 0.1309 | <0.0001 |
| Incidence L0-14 | 3.41 | 2.46 | 4.13 | 9.70 | <0.0001 | <0.0001 |
| Incidence adjusted | 3.36 | 1.93 | 4.30 | 7.55 | <0.0001 | <0.0001 |

Note:

\* No statistical significance for the subtropical lag of 7 days; however, $p$ of the lag of 14 days is <0.0001.

In Figs. 6E and 6F, the dose–response relationships of daily means of relative humidity remained positive for the (sub)tropical climates, even after adjusting the biased slope. On Figs. 6G and 6H, the bivariate smoothing of the daily means of temperature and relative humidity by the LOESS model also revealed the inverse relationships for (sub)tropical climates.

Each 1 °C rise of the mean daily adjusted temperature was associated with a −6.89% ($t = -2.99$, $p = 0.0029$) decrease for subtropical climate and a 12.14% ($t = 12.35$, $p < 0.0001$) increase in the incidence of COVID-19 for the tropical climate. Each 1% rise of the means daily adjusted relative humidity was associated with a 2.35% ($t = 4.17$, $p < 0.0001$) and a 2.26% ($t = 8.88$, $p < 0.0001$) increase of COVID-19 cases for both (sub)tropical climates, respectively.

Tables 3 and 4 summarize the effects of 1 °C and 1% increases of daily temperature and relative humidity, respectively, within the analyses.

Consistent results of the sensitivity analyses yielded robust findings also including the lag of days. As stated in Zhu et al. (2020), the incubation period of COVID-19 combined with the delay of Sars-CoV-2 testing results necessitated a moving-average approach to account for the cumulative lag effect of temperature and relative humidity.

## DISCUSSION

The lack of scientific evidence on the environmental proliferation of COVID-19 in tropical countries prompted this investigation of the correlations between climatic variables and infectivity rates of SARS-CoV-2 through a Generalized Additive Model (GAM) and the Locally Estimated Scatterplot Smoothing (LOESS) nonparametric model. Approximately 83% of testing has been conducted in non-tropical countries (30° N and higher). Likewise, ~90% of COVID-19 cases have been recorded in the same countries within a temperature range of 3–17 °C.

Studies have shown a significant inverse relationship between temperature and relative humidity with the spread of SARS-CoV-2 (*Bukhari & Jameel, 2020*; *Liu et al., 2020*; *Zhu et al., 2020*; *Núñez-Delgado, 2020*; *Yip et al., 2007*; *Thai et al., 2015*; *Ng, Basta & Cowling, 2014*; *Lowen & Steel, 2014*; *Bi, Wang & Hiller, 2007*; *Barreca & Shimshack, 2012*; *Moriyama & Ichinohe, 2019*; *Casanova et al., 2010*; *Wang, Goggins & Chan, 2018*; *Xu et al., 2020*; *Daltio et al., 2018*; *Chan et al., 2011*; *Van Doremalen, Bushmaker & Munster, 2013*). Recent studies have shown that SARS-CoV-2 has the highest predicted half-life at 30 °C after drying, and virus titers remained more stable there over time than they did at lower temperatures (*Kratzel et al., 2020*). Another recent experiment revealed that the likelihood of survival for SARS-CoV-2 is roughly five times higher in humid conditions as it is in dry conditions (*Bhardwaj & Agrawal, 2020*). These studies are consistent with the spread of COVID-19 in Brazil (*Auler et al., 2020*).

Although respiratory droplets seem to play a central role in SARS-CoV-2 transmission (*Anfinrud et al., 2020*), surface contact transmission has also been proven to have a significant impact on the spread of the virus (*Casanova et al., 2010*; *Anfinrud et al., 2020*). In addition, temperature and humidity variations influence SARS-CoV-2 stability and survival (*Duan et al., 2003*). The virus can remain active on surfaces from a few hours to several days, depending on these conditions (*Van Doremalen, Bushmaker & Munster, 2013*).

The research findings by the use of Köppen climate types helped us to understand to what extent the seasonal changes associated with climatic factors fluctuations might actively influence virus survival. Köppen climate types are major classified by temperature and the distinction between wet and dry climates. Until now, most of the cited studies are in temperate zones where the virus found stability in dry and cold regions. This study showed that the virus also find stability in hot and wet zones, like the tropical climate. It was possible because of the territorial magnitude of Brazil, which include tropical, hot and wet, and subtropical, milder and dry, climates.

The observed data results have some limitations: other factors might influence findings, like population density, virus resistance, population mobility and endurance, individual health factors, and so on. While much about the virus remains a mystery, the growing body of research shows a need for public safety measures to curtail its spread. These include social distancing, wearing masks, using hand sanitizer, and quarantining, all of which may have directly impacted the findings of this study. All of the Brazilian state

capitals have at least employed some kind of social distancing, but the long-term efficacy of these measures will require further study.

## CONCLUSION

Findings revealed an inverse relationship between subtropical and tropical climates for the spread of the novel coronavirus and temperature, suggesting a sensitivity behavior to climates zones, and perhaps hemispheres. Indeed, Köppen climate types do not follow latitudinal order to define climate zones. Outcomes conjecture a temperature pattern for the spread of COVID-19 for (sub)tropical climates, at least in Brazil, and may contribute to clarify the role of environmental factors for the spread of the novel coronavirus for different climate zones. Future climatic model studies should include solar variables such as geomagnetic or solar radiation, which have planetary incidence with space weather.

### Funding

The Ministry of Health of Brazil supported this study (TED 43/2019). The funders had no role in study design, data collection and analysis, decision to publish, or preparation of the manuscript.

### Grant Disclosures

The following grant information was disclosed by the authors:
The Ministry of Health of Brazil: TED 43/2019.

### Competing Interests

The authors declare that they have no competing interests.

### Author Contributions

- David Prata conceived and designed the experiments, analyzed the data, prepared figures and/or tables, and approved the final draft.
- Waldecy Rodrigues conceived and designed the experiments, analyzed the data, authored or reviewed drafts of the paper, and approved the final draft.
- Paulo Henrique De Souza Bermejo conceived and designed the experiments, prepared figures and/or tables, authored or reviewed drafts of the paper, and approved the final draft.
- Marina Moreira performed the experiments, authored or reviewed drafts of the paper, and approved the final draft.
- Wainesten Camargo performed the experiments, authored or reviewed drafts of the paper, and approved the final draft.
- Marcelo Lisboa performed the experiments, prepared figures and/or tables, and approved the final draft.
- Geovane Rossone Reis performed the experiments, authored or reviewed drafts of the paper, and approved the final draft.
- Humberto Xavier de Araujo performed the experiments, prepared figures and/or tables, and approved the final draft.

## Data Availability

Raw data are available as a Supplemental File.

## Supplemental Information

Supplemental information for this article can be found online at http://dx.doi.org/10.7717/peerj.10655#supplemental-information.

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
