# Peer review of "The relationship between (sub)tropical climates and the incidence of COVID-19"

_PeerJ, doi:10.7717/peerj.10655_

## Round 0.1 · original submission · Major Revisions

Dear author, we have reviewed your manuscript and, considering the importance of this type of study, I consider it appropriate to make improvements, especially considering those of the first author. On the other hand, Line 181-202 as mentioned in the lack of evidence, I consider it appropriate to mention that perhaps in future studies the model should include solar variables such as geomagnetics or solar radiation that have planetary incidence with space weather.

·

Basic reporting

However, I have some comments to improve the quality of the document
In the abstract, it is necessary a better description of materials and methods, as well as the most important results, the authors too much emphasis on justification, but not enough mention is made of the methodology.
The Authors write This study intends to corroborate with both laboratory experiments and real-world data to provide rationale for the adoption of public health policies. What laboratory experiments are they referring to? to covid tests? or maybe to statistical models?
This is not described in detail in the methodology
Introduction
Line 50, after 2019 space, the same for all documents
For editor,

Line 53, citations are not in the correct format, the same for all documents

Experimental design

In general, the methodology requires a more detailed description
Line 94: describe what the acronym INMET means

Validity of the findings

The results are tainted with arguments and assumptions
Line 124. Why do the authors report Spearman and Pearson correlation? should be one of the two based on the parametric distribution of the numerical data. In addition, the Spearman correlation between COVID incidence and mean temperature is negative (-0.08075), not positive.
Line 140-142, remove “The incubation period of COVID-19 combined with the delay of Sars-CoV-2 testing results necessitated a moving-average approach to account for the cumulative lag effect of temperature and relative humidity[7].” gives the impression of being discussion and not results

In table 1 you must put what T and RH and N (observation?) Mean
Figure two has poor resolution, it is impossible to read

Additional comments

The Study is relevant, mainly taking into consideration what was stated by the authors regarding that in tropical areas, people have greater contact with the external environment, the results of studies of this type, without a doubt, are of importance for decision-making and the implementation of local policies for infection prevention.

Reviewer 2 ·

Basic reporting

no comment

Experimental design

The knowledge gap being investigated should be clearly mentioned and method should have been described in technical high standards.

Validity of the findings

The findings figure should have been simplified. In present form it contains lot of data and difficult understand from the layman's perspective. This is needed as today "covid-19" topic is related to every member of society. Authors can add/present their conclusive figure and in which their novelty aspect can be highlighted.

Additional comments

In all figures, there is overlapping of lot of data. Too much scattering of data is difficult to bring out the novelty or authors contribution in this work. I would suggest add one extra figure, it could be a schematic based highlighting the contribution.

---

## Round 0.2 · Minor Revisions

On the second discussion, is neccesary to make some improvenmentes, so the improvements are minor and form, I would greatly appreciate making the observations indicated by the reviewers ....... Clear and aware of the importance of this type of study in the world situation, I hope your improvements soon.

·

Basic reporting

I can actually notice a great improvement over the first document
I just have a few little suggestions
In the abstract the wording was improved, however, the results still need to be better described, the results of the correlations can be described

Experimental design

A more detailed description of the methodology was made, Figure 2 helps to better understand the study design

Validity of the findings

In tables 2, 3, and 4 delete the columns, similar to table 1
In the figures in general the y-axis describe the incidence that is used (per 100k habitants?), Similarly in the X-axes, ° C should be added to T, that is, the units of measurements

Additional comments

I'm glad the necessary changes have been made

Reviewer 2 ·

Basic reporting

no comment

Experimental design

Happy to see the revised text and the added figure 2.

Validity of the findings

No comment

Additional comments

No comment

Reviewer 3 ·

Basic reporting

See attachment.

Experimental design

See attachment.

Validity of the findings

See attachment.

Annotated reviews are not available for download in order to protect the identity of reviewers who chose to remain anonymous.

---

## Round 0.3 · accepted · Accept

Dear thanks for your incorporated improvements and clarifications so your manuscript is ready for publication. Thanks again for your contribution to the COVID-19 pandemic.